# The Assessability of Approximal Secondary Caries of Non-Invasive 3D-Printed Veneers Depending on the Restoration Thickness—An In Vitro Study

**DOI:** 10.3390/bioengineering10090992

**Published:** 2023-08-22

**Authors:** Elisabeth Prause, Jeremias Hey, Franziska Schmidt, Robert Nicic, Florian Beuer, Alexey Unkovskiy

**Affiliations:** 1Department of Prosthodontics, Geriatric Dentistry and Craniomandibular Disorders, Charité-Universitätsmedizin Berlin, Corporate Member of Freie Universität Berlin and Humboldt-Universität zu Berlin, Aßmannshauser Str. 4-6, 14197 Berlin, Germany; jeremias.hey@uk-halle.de (J.H.); franziska.schmidt2@charite.de (F.S.); robert.nicic@charite.de (R.N.); florian.beuer@charite.de (F.B.); alexey.unkovskiy@charite.de (A.U.); 2Department of Prosthodontics, School of Dental Medicine, Martin-Luther-University, 06112 Halle, Germany; 3Department of Dental Surgery, Sechenov First Moscow State Medical University, Bolshaya Pirogovskaya Street, 19c1, 119146 Moscow, Russia

**Keywords:** 3D printing, additive manufacturing, non-invasive, secondary caries, veneers

## Abstract

To date, no scientific data is available regarding the development and radiographic assessment of approximal caries development after the insertion of 3D-printed, non-invasive veneers of different restoration thicknesses. For the present study, non-invasive veneers were fabricated from two different materials for printing and milling (Vita Enamic and VarseoSmile Crown plus). Three different restoration thicknesses (0.5, 0.7, and 0.9 mm) were selected. After digital design, leaving the approximal space free, and manufacturing of the restorations, adhesive insertion followed. All specimens were placed in a demineralizing solution for 28 days. Subsequently, a radiological and fluorescent examination was performed. The present study showed statistically significant interactions for the day (*p* < 0.0001) and manufacturing method (*p* < 0.0001) but not for restoration thickness. Additive manufactured restorations showed less radiological caries progression compared to subtractive manufactured restorations after 21 and 28 days (0.7 and 0.9 mm restoration thickness) (*p* < 0.0001). DIAGNOdent proved that the restoration thickness affected the caries progression within the subtractive group (*p* < 0.0001). Radiographic and fluorescence examination showed equivalent results regarding approximal caries assessment. For additive manufacturing, less caries progression was shown without consideration of the restoration thickness.

## 1. Introduction

Digital dentistry is increasingly coming into focus. A digital workflow using computer-aided design/computer-aided manufacturing (CAD/CAM) has yielded various new hybrid materials for subtractive manufacturing (SM). They are characterized by improved mechanical and esthetic properties [1,2,3,4].

However, SM is considered time-consuming and inefficient regarding material consumption. Furthermore, delicate marginal areas cannot be realized by milling. In contrast to SM processes, additive manufacturing (AM) offers a more economical material consumption, while multiple and complex restoration geometries can be produced simultaneously [5,6,7], reducing manufacturing time and costs [8]. In addition to millable hybrid materials, printable dental materials are increasingly coming to the fore. One CAD/CAM printable hybrid material (VarseoSmile Crown plus, Bego, Bremen, Germany) is approved for permanent single-tooth restorations according to the Medical Device Regulation as class IIa material. Other printable materials are approved only for temporary restorations. The material promises excellent mechanical properties, a good fit for non-invasive restorations, low material consumption, and inexpensive production. As far as this material allows for manufacturing very thin (up to 0.3 mm) restorations, 3D-printed restorations on single teeth could be applied in a less invasive or non-invasive way to protect sound dental hard tissues. In the case of a non-invasive approach, the approximal areas are not separated from each other and cannot be covered by a restoration. Furthermore, the restoration margin is often extended close to the approximal area, making it less attainable for dental hygiene and producing a predisposed area for caries. Consistent oral hygiene and regular dental and radiological examinations would be indispensable in a non-invasive approach.

Excessive tooth wear due to erosion, attrition, and abrasion usually leads to the need for restorative measures [9]. Based on an increasing number of patients suffering from erosions, attrition, and abrasion, complex prosthetic rehabilitations occur more often. If extensive areas of the enamel or dentin are affected, hypersensitivities, deteriorated esthetics, and a decrease of the vertical dimension of occlusion (VDO) might occur, for which reestablishment is a time-consuming process [9,10]. The conservative treatment is invasive and expensive for the patient. With the help of non-invasive, 3D-printed restorations, a rehabilitation of the VDO could be achieved, preserving the existing dental hard tissues. Also, the treatment time and costs could be reduced. Using a completely digital workflow, complex prosthetic rehabilitations could become easier, cheaper, and less invasive than the conventional treatment concept. Esthetic improvements could be achieved immediately. So far, no in vivo data regarding 3D-printed, non-invasive restorations are available.

For a reliable dental radiological examination, an adequate radiopacity of a dental material is crucial for assessing the marginal integrity of the restoration, detecting gaps in the interface, diagnosing secondary caries, and distinguishing the material from dental hard tissues [1,11,12,13,14]. A radiopacity of a material equal to or higher than that of dentine is considered beneficial for dental diagnostics [1,15]. It is known that radiopacity values depend on the materials’ filler content and quantities, for example, the glass, ceramic, and/or resin filler content [1]. Therefore, for the clinical suitability of a new dental restorative material, the evaluation of the radiopacity is necessary concerning the respective indication area of the material [1,16,17].

In contrast to the advantages of a higher radiopacity described above, a low radiopacity could bring advantages in assessing the approximal areas after insertion of non-invasive 3D-printed restorations with regard to the development of caries. So far, no scientific data on the radiopacity of non-invasive 3D-printed restorations and approximal caries detection is available. Whether and when an approximal lesion is restored depends on its progression [18]. Clinically, close radiological monitoring is recommended [18]. Also, other diagnostic tools for caries detection are available. Based on fluorescence, caries development can be assessed and compared to radiographic examination. Nowadays, remineralizing procedures are preferred instead of restorative treatment [18,19,20].

Previous studies outside of dentistry already used X-ray analysis in order to draw conclusions about the microstructure of 3D-printed materials [21,22,23]. The microstructure gives information about the porosity and mechanical properties of a material, particularly the amount and distribution of voids within the intra- and inter-layer structure of a 3D-printed material [21]. Furthermore, the filler distribution after the printing process is of great interest for the final restorations. So far, no data are available regarding 3D-printed dental restorative materials. Therefore, the present study could provide the first data regarding X-ray depiction and analysis of a 3D-printed dental material.

The aim of the present study was to assess radiographically and with the help of fluorescence, the incidence, and progression of approximal caries in the enamel after insertion of non-invasive veneers of different restoration thicknesses and different manufacturing processes on the 7th, 14th, 21st, and 28th day. The working hypothesis was that restoration thickness, manufacturing method, and day do not influence the caries lesions assessment by non-invasive restoration type.

## 2. Materials and Methods

### 2.1. Specimen Preparation

Twenty-four human incisors were collected and provided by the oral surgery department of the Charité-Universitätsmedizin Berlin for the present study. The teeth were randomly divided into three groups (*n* = 8) according to the different tested restoration thickness of the non-invasive restorations (0.5, 0.7, and 0.9 mm) (Figure 1 and Figure 2).

All teeth were scanned using an intraoral scanner (Primescan, Dentsply Sirona, Charlotte, NC, USA). The data were transmitted to the dental laboratory of the Charité. Here, the digital design, excluding the approximal areas of each tooth, and the manufacturing process of the non-invasive veneers were carried out. For the additive group, the printable material (VarseoSmile Crown plus, Bego; printer: Varseo XS, Bego), and for the subtractive group, the millable material (Vita Enamic, Vita Zahnfabrik, Bad Sackingen, Germany) (Table 1) were used to manufacture the restorations. Fabrication and postprocessing were carried out according to the manufacturer’s instructions.

For bonding the restorations on the specimens, all teeth were etched with 37% phosphoric acid for 30 s (Ätzgel 37%, Orbis Dental Handel mbH, Münster, Germany). The etching gel was removed thoroughly with a dental sprayer, and the teeth were dried with compressed air for 30 s. Afterward, a self-etching universal adhesive (Scotchbond SE, 3M Espe, Landsberg am Lech, Germany) was applied on the surface of each tooth. The restorations were cut in the middle with the help of a cutting disk. Consequently, a direct radiographic comparison between a covered and an uncovered half of each tooth could be drawn. The restorations were air-abraded with aluminum oxide and conditioned with a universal primer (Monobond Plus, Ivoclar, Schaan, Liechtenstein). A luting composite (RelyX Ultimate, 3M Espe) was used for fixation. After the removal of the cement residues in all other areas, light curing was conducted. Subsequently, all uncovered areas of the enamel, except for the proximal areas, were coated with nail varnish (Manhattan Super Gel, Paris, France) (Figure 3a,b).

### 2.2. Artificial Carious Lesions

To create artificial carious lesions in the mesial and distal proximal areas of each tooth, the specimens were exposed to a demineralizing solution for 28 days (pH 4.95; 37 °C) [24]. The pH was checked daily and, if necessary, corrected with a potassium hydroxide solution (10 M) [25]. The specimens remained in the demineralizing solution for a total of four weeks.

### 2.3. Radiological and Fluorescence Examination

For a reproducible and reliable radiological examination, all teeth were embedded in a specially prepared and fixed X-ray holder (Luxatemp Automix Plus, DMG Chemisch-Pharmazeutische Fabrik GmbH, Hamburg, Germany) (Figure 4). It allowed the specimen to be rotated by 45° in each case so that mesial and distal eccentric radiographs could be taken in addition to an orthoradial image. The distance between the head of the X-ray unit and the specimen was 8 cm. All teeth were radiologically checked in advance to ensure that they were free of caries and restorations approximally (Figure 5a–f). All specimens were examined radiologically at intervals of 7 days with regard to the development of a carious lesion in the approximal areas depending on the different restoration thicknesses and manufacturing methods tested. An X-ray voltage of 70 kV and an energy of 19.6 mGy·cm^2^ was used (Heliodent plus, Dentsply Sirona, Charlotte, NC, USA). The radiographs were processed immediately in an automatic processor (Vista Scan, Dürr Dental, Bietigheim-Bissingen, Germany).

Additionally, approximal caries detection was conducted with the help of DIAGNOdent 2095 (KaVo Dental GmbH, Berlin, Germany). The DIAGNOdent scores range between 0 and 99. In the present study DIAGNOdent values < 15 represented no demineralization. Values between 15 and 19 demineralization extended into the inner half of enamel up to the upper third of dentin. Values <19 represented a demineralization extending to deeper dentin [26] (Figure 6).

### 2.4. Data Analyses

The evaluation of the radiographs and DIAGNOdent analysis of all specimens was conducted in accordance with the studies by Mejáre et al. [18]. The following scoring systems were used regarding an evaluation of the approximal areas of each tooth (Table 2; Figure 7). Two clinicians conducted the analysis (E.P. and A.U.). For calibration purposes, ten radiographs (orthoradial, mesial- and distaleccentric) were analyzed and discussed in common [18]. About two-thirds of all X-ray images were then analyzed by one clinician (E.P.). The rest of the X-ray images were analyzed by the other (A.U.). To perform an intra-examiner reproducibility, 10% of all radiographs were analyzed twice by each clinician mentioned above. For inter-examiner reproducibility, another 15% were analyzed by both. The radiographs were selected randomly by an independent assistant. Consequently, it could be ensured that the clinicians did not analyze the same radiograph for a second time. The reproducibility was tested and calculated in accordance with the κ-values for diagnoses published by Cohen et al., 1960 [27]. The inter-examiner agreement had a κ-value of 0.57, and the intra-examiner agreement showed κ-values of 0.79 and 0.72.

### 2.5. Statistical Analyses

The gathered data was expressed as mean and standard deviation. A two-way analysis of variance (ANOVA) was performed to evaluate the statistically significant differences, with the restoration thickness and material as two independent factors. The gathered data was analyzed for the goodness-of-fit using the Shapiro–Wilk test. Tukey’s multiple comparisons tests were used for multiple comparisons analyses. All statistical analyses were performed with JMP 14 software (SAS Corp., Heidelberg, Germany). A *p*-value less than 0.05 was defined as statistically significant.

## 3. Results

The radiological examination revealed statistically significant interactions for the day (F = 13.3; *p* < 0.0001) and manufacturing method (F = 11.4; *p* < 0.0001) but not for restoration thickness (F = 2.38; *p* < 0.09).

Figure 8 illustrates the post-hoc multiple comparisons between all the groups. The overall tendency indicated that the additive group showed less radiological caries progression than the subtractive for the 0.7 and 0.9 mm restoration thickness on the 21st and 28th day (*p* < 0.0001). There was no statistically significant difference with regard to restoration thickness within the additive group. However, in the subtractive group, the 0.7 and 0.9 thicknesses demonstrated more radiological caries progression on both the 21st and 28th days (*p* < 0.0001).

The DIAGNOdent method proved the fact that the restoration thickness affected the caries progression within the subtractive group (*p* < 0.0001) (Figure 9). No difference in caries progression was observed within the additive group.

## 4. Discussion

In the present study, different effects of non-invasive veneers of different restoration thicknesses and manufacturing methods regarding approximal caries development could be shown. A significant correlation between the days and the manufacturing process (additive) could be proven. The restoration thickness had a significant influence on milled, non-invasive veneers. The working hypothesis that restoration thickness, manufacturing method, and the day do not influence the caries lesions assessment by non-invasive restorations type was therefore rejected.

The working hypothesis that restoration thickness, manufacturing method, and the day do not influence the caries lesions assessment by non-invasive restorations type was therefore rejected.

In the present study, a direct light processing (DLP) 3D printer was utilized for additive manufacturing of non-invasive restorations. There have been some studies reporting the stereolithography (SLA) method for manufacturing such composite–ceramic hybrid restorations [28,29]. Whether the SLA-printed restoration would demonstrate the same caries rate and its assessability should be examined in further research.

Until now, additive-manufactured restorations in fixed prosthodontics are new and mainly unexplored [30]. The accuracy of 3D-printed restorations was evaluated positively in the still-reduced data situation. Studies showed marginal and internal fits that were superior to those of milled restorations [30,31,32]. However, there is only limited data on the clinical performance of 3D-printed restorations, and such aspects as caries and marginal integration over a certain period of intraoral use are not yet evaluated. For this reason, the present study aimed to describe the influence of printed and milled non-invasive restorations on approximal caries development. These data should be considered as the first classification of the material. In the sensitive approximal region, additive manufacturing could offer advantages in the future with regard to the prevention of secondary caries.

The different principles of light projection between DLP and SLA may have some influence on the translucency of restorations and, therefore, on the radiological detectability of caries. In addition, the materials processed by DLP and SLA methods differ in their chemical composition because of the photosensible pigments, which may also result in an alternative radiopacity. Furthermore, the restoration thickness and layer orientation may also influence the internal structure of restorations. In the present study, the 50 µm restoration thickness and 45° build angle have been used. The restorations have been glazed prior to the fixations, which added some thickness and might have also influenced the optical properties of restoration. This effect may be neglected in the case of mechanical polishing instead of glazing. Thus, the avenue of materials choice and additive manufacturing method and restorations post-processing must be pursued in future research.

Based on the results of the present study, it was shown that the manufacturing process had a significant influence on caries detection. However, this could be due to ultra-thin printable restoration thicknesses and the resulting positive marginal adaptation. In this trial, non-invasive veneers were manufactured. So far, permanent 3D-printed single-tooth restorations have only been approved by the FDA (Food and Drug Administration) as grade IIa materials for prepared teeth [30]. However, in the long term, non-invasive 3D-printed restorations will provide a real therapeutic gain since printing ultra-thin restoration thicknesses are the real advantage compared to milled restorations. However, there is limited literature available on the accuracy of 3D-printed restorations for both invasive and non-invasive restorations. Furthermore, clinical data does not exist so far.

Regarding translucency, scientific data about 3D-printable materials is again scarce. Due to its liquid character, the distribution of fillers might be inhomogeneous. The filler content of the tested printed material (VSCP) is set between 30–50% by the manufacturer. To achieve optimal 3D-printing outcomes, a more flowable consistency of a pre-polymerized/raw composite resin, which has a reduced amount of inorganic filler content of the resin composite, is needed. However, the absence or decreasing amount of inorganic filler lowers the mechanical properties of 3D-printed resin or composite resin, narrowing their clinical indications to long-term interim restorations [33]. Milled CAD/CAM blocks seem to offer a more homogeneous microstructure since they are manufactured industrially. The microstructure of the tested milled material (VE) is described as a polymer-infiltrated ceramic network with a filler content of 86%, according to the manufacturer. Variations in translucency were related to differences in crystal volume and the scattering of light regarding all-ceramic materials [34]. Less scattering of light can be achieved by a less crystalline content. Therefore, the translucency can be influenced by the crystal volume [34]. However, a lower translucency could be beneficial for caries detection regarding CAD/CAM hybrid materials for milling and printing. 3D-printed, non-invasive veneers could offer an easier and more predictable caries assessment due to a lower translucency. Furthermore, no radiological data about the microstructure of 3D-printed dental restorations are available today. It is known that the printing process, as well as the composition of the material, can have a great impact on mechanical properties. Scientific data regarding industrial materials and 3D printing of silicones exist [21,22,23]. Studies showed that a higher porosity was found in the interlayer regions [21]. Consequently, mechanical strength could be reduced. Higher porosity and inhomogeneous distribution of the ingredients affect the material properties [21,35,36]. A precise analysis of the microstructure of the 3D-printed material using X-ray has been used in previous studies as well [21,37] to derive information regarding the pore shape, size, distribution, orientation, and position [21] of the ingredients. Computed tomography (CT) has been used for an investigation of additive manufactured materials since a 3D analysis of the tested material or structure became possible [22]. It is considered the most effective nondestructive test method for measuring the internal features of a 3D-printed material [22,38]. Since a volumetric analysis has become possible, an evaluation of the 3D details of pores, and their shape, density, and distribution, for example, in 3D-printed metallic materials, could be conducted with the help of CT. However, regarding approximal caries detection in dentistry, two-dimensional X-ray is considered the gold standard. Therefore, it has been used in the present study as well.

However, further in-vivo studies are necessary. As for now, the composite–ceramic hybrid materials are the only ones to be used for a non-invasive thin restoration. However, there have been clinical reports on ceramics 3D printing. The clinical approvement for the 3D-printed ceramics will open new horizons for the investigation of caries detectability by a 3D-printed non-invasive restoration.

The application of DIAGNOdent showed a positive correlation with radiographic examination in the present study. The technology is based on laser fluorescence, which measures the different laser fluorescence of healthy and carious dental hard tissues [39]. The advantage is the absence of ionizing radiation [39]. Previous study results showed high accuracy and capability of DIAGNOdent with regard to caries detection [39,40,41,42,43,44]. However, only one study has so far demonstrated a significant difference [39,40]. Several studies indicated that DIAGNOdent should be used as an additional diagnostic tool [39,45,46]. The detection of approximal secondary caries, as in the present study, is more complicated than the detection of primary caries [39]. So far, different diagnostic tools for caries detection have been applied and investigated. These include digital and analog radiography, but also laser fluorescence [39,47,48,49]. The evaluation of radiographs requires clinical experience [39]. DIAGNOdent as a diagnostic tool could, therefore, also be useful for less experienced users, provided that the high accuracy is proven [39,50]. The results of the present study should be used for further studies with a direct comparison with radiographs so that a reliable clinical application can be guaranteed [39].

The present study presents a radiological and fluorescence analysis of a novel CAD/CAM 3D-printable hybrid material. A direct comparison to an established CAD/CAM millable hybrid material was drawn. Such studies are essential to provide data regarding the material microstructure based on different manufacturing processes, yet they are currently lacking. One main limitation of our study is that only one 3D printer and one printing strategy were used. Furthermore, the testing of different CAD/CAM hybrid materials for milling and printing would have been desirable. Also, in vivo data would provide valuable information regarding material properties. Little is known about the long-term clinical behavior of 3D-printed restorations compared to milled restorations. CAD/CAM technology, as well as the development of high-density polymers based on highly cross-linked polymethylmetacrylate (PMMA) or newly available CAD/CAM printable hybrid materials, offer new clinical and technical options for complex prosthetic rehabilitations. Advancements in CAD/CAM technology have facilitated the integration of digital workflow into clinical treatment sequences [5]. In addition to subtractive manufacturing of restorations, additive manufacturing is increasingly coming to the fore. Nowadays, 3D-printed restorations on single teeth can be applied in a non-invasive procedure. By using an intraoral scan and a digital facebow [51,52], the status quo of the occlusion and maxillomandibular relationship could be recorded. The data can afterward be uploaded and integrated into the virtual articulator in the design software. No pretreatment with a splint is necessary. Esthetic improvements are achieved immediately. Hypersensitivities can be eliminated. Preparation of the teeth is not necessary. The already reduced dental hard tissues can be preserved. Compared to CAD/CAM millable hybrid restorations, CAD/CAM printable hybrid restorations offer more delicate marginal areas. AM offers the possibility of realizing very thin layer thicknesses for non-invasive restorations compared to milling. However, the treatment concept using 3D-printed restorations has not been evaluated in an in vivo environment until now. Further in vitro and in vivo studies are required to collect sufficient data and knowledge about these novel available dental materials based on a complete digital workflow.

## 5. Conclusions

The present study revealed that both radiographic and fluorescence examinations were able to produce equivalent results regarding approximal caries development assessment. For additive manufacturing, less caries progression without consideration of the restoration thickness could be shown as in the subtractive method. For subtractive manufacturing, caries development after 21 and 28 days by 0.7 and 0.9 mm restoration thickness could significantly be proven. DIAGNOdent and radiography showed a positive correlation. This preliminary in vitro analysis sheds some light on the clinical aspects of 3D-printed restorations application and demonstrates that additive manufacturing of such restorations may compete with subtractive ones. Surely, long-term preclinical and clinical data is needed to assess the performance of non-invasive 3D-printed thin restorations.

## Figures and Tables

**Figure 1 bioengineering-10-00992-f001:**
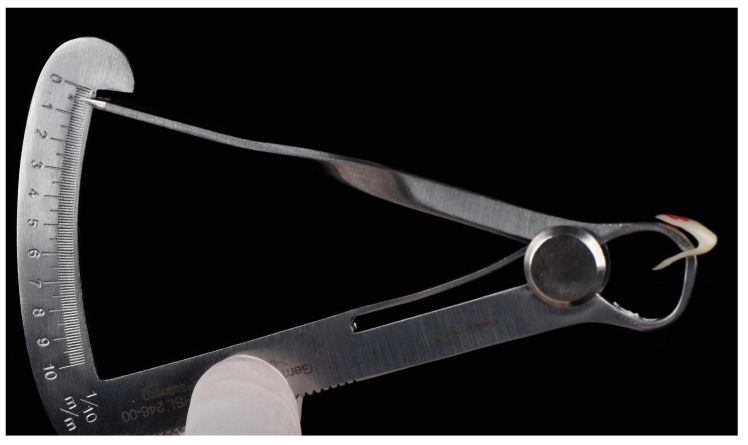
Depiction of the restoration thickness (0.5 mm) of one 3D-printed, non-invasive Veneer.

**Figure 2 bioengineering-10-00992-f002:**
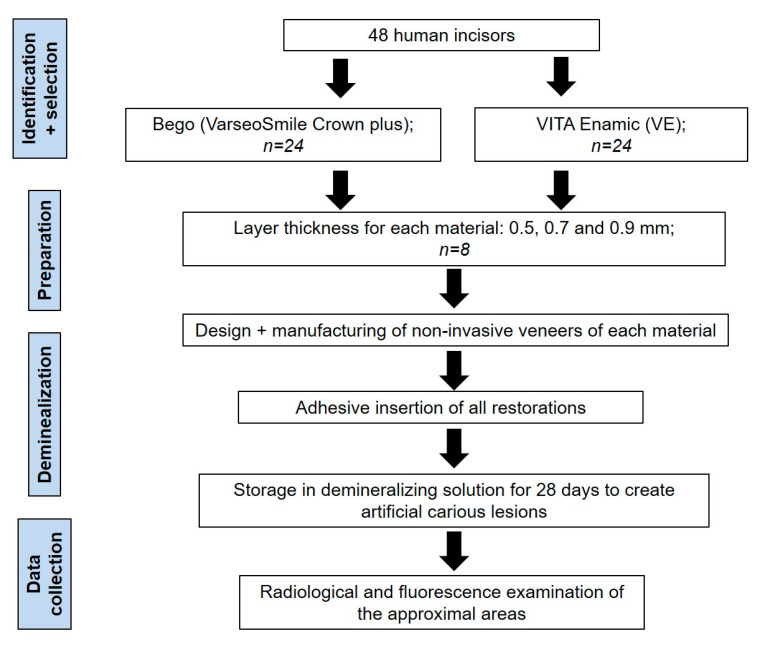
Schematic of the workflow of the entire experiment.

**Figure 3 bioengineering-10-00992-f003:**
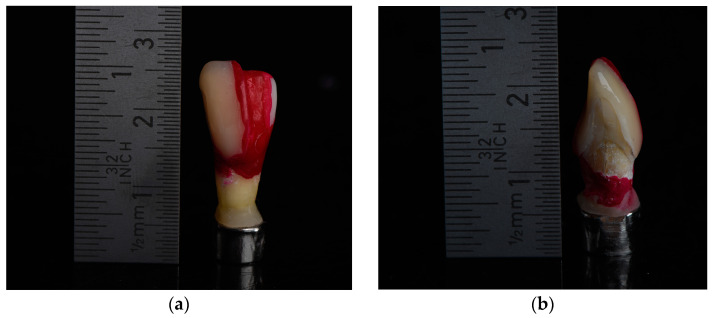
(**a**,**b**) Embedded specimen with an already bonded 3D-printed restoration on half of the tooth and prepared with nail varnish covering before placement in demineralizing solution. The approximal areas were not covered with nail varnish in order to expose these areas to the demineralizing solution.

**Figure 4 bioengineering-10-00992-f004:**
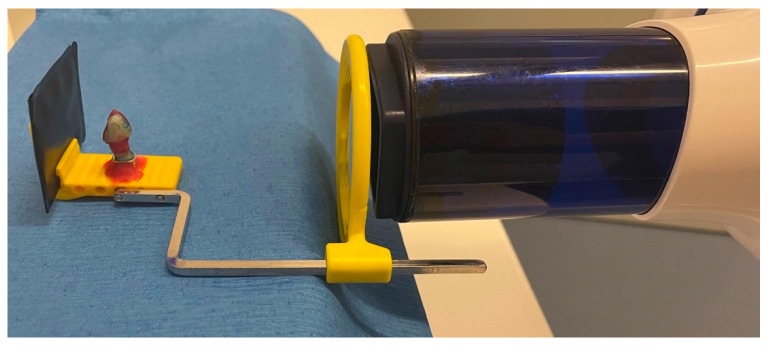
Placement of the specimen in the X-ray holder with a defined distance to the X-ray source.

**Figure 5 bioengineering-10-00992-f005:**
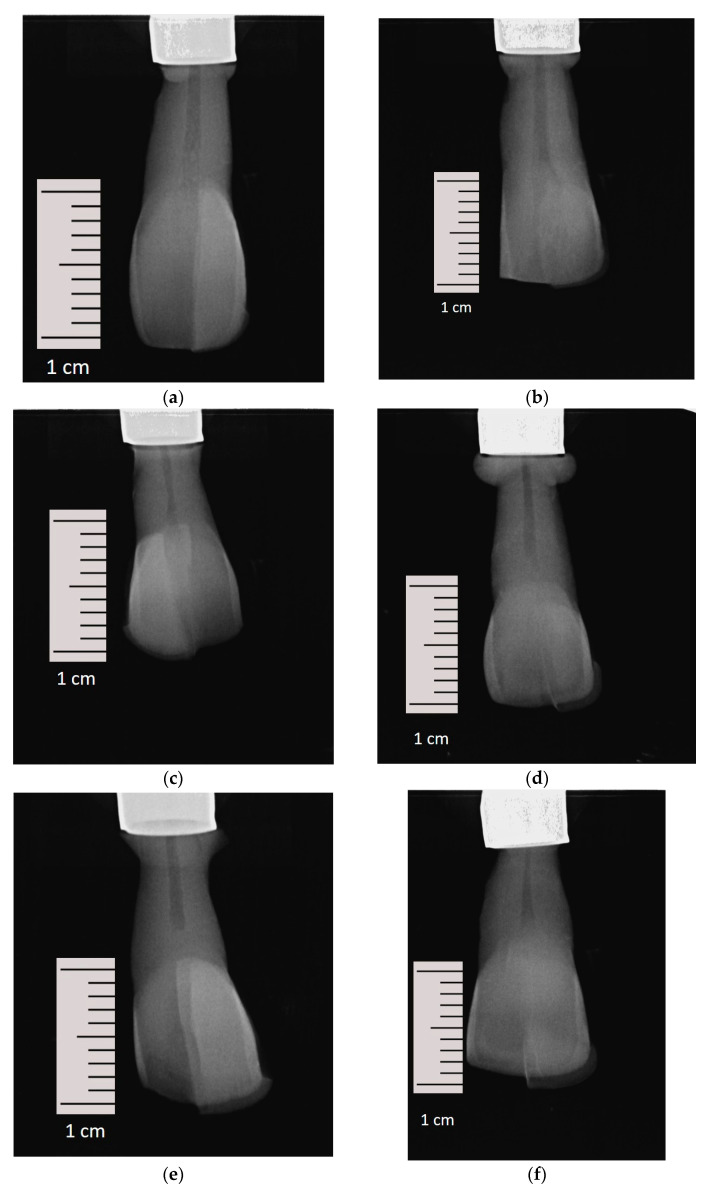
(**a**–**f**). Initial orthoradial X-ray of one sample restored with VSCP ((**a**) = 0.5 mm), ((**c**) = 0.7 mm), ((**e**) = 0.9 mm) and VE ((**b**) = 0.5 mm), ((**d**) = 0.7 mm), ((**f**) = 0.9 mm) with different layer thicknesses.

**Figure 6 bioengineering-10-00992-f006:**
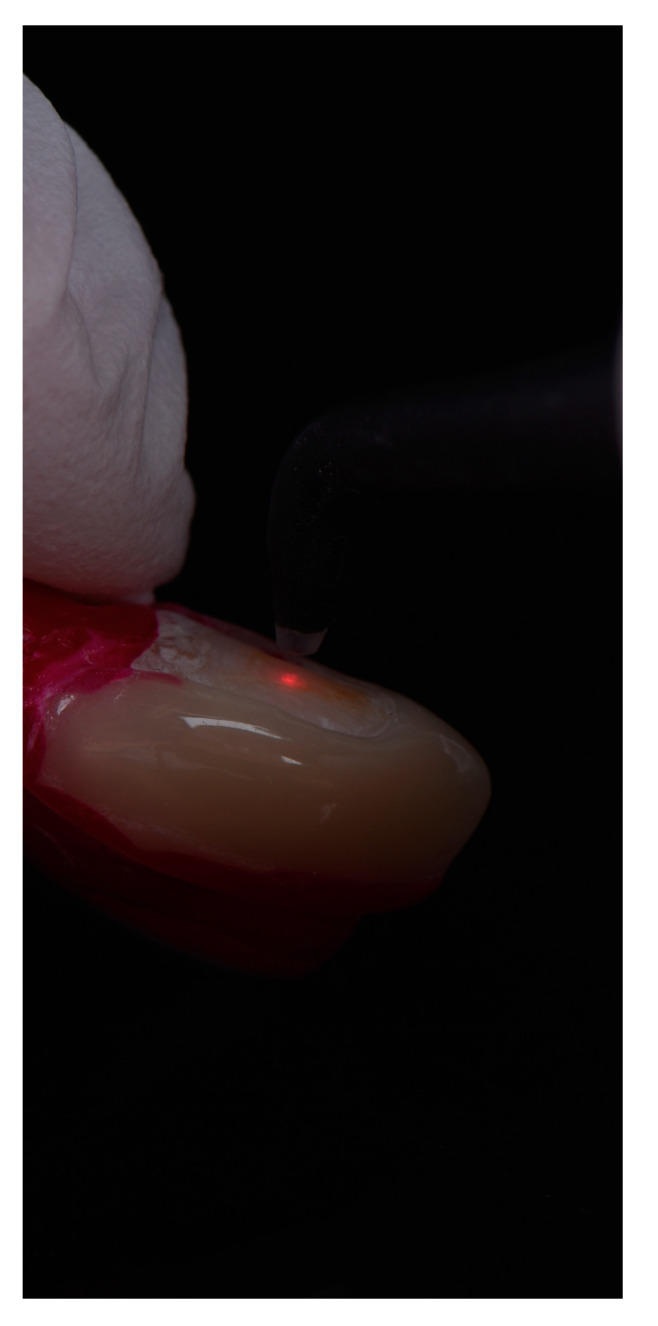
Laserfluorescence measurement on an approximal space.

**Figure 7 bioengineering-10-00992-f007:**
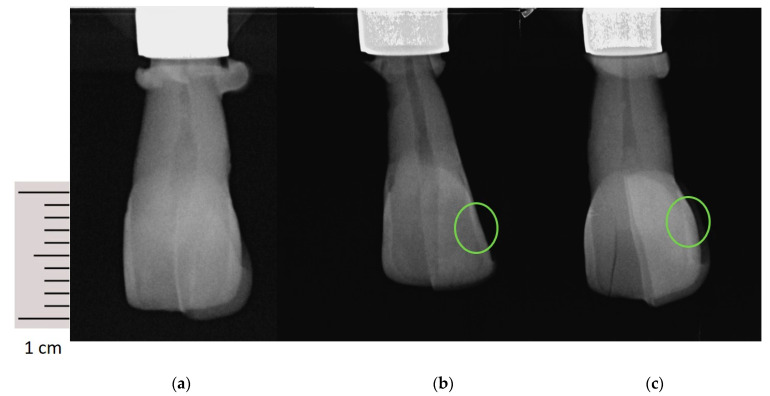
(**a**) Initial orthoradial X-ray of one sample restored with VSCP (Score 0); (**b**) Orthoradial X-ray restored with VSCP (Score 1) after 28 days of demineralization; (**c**) Orthoradial X-ray restored with VSCP (Score 2) after 28 days of demineralization. The green circles mark the developed artificial caries lesions of different scores over time.

**Figure 8 bioengineering-10-00992-f008:**
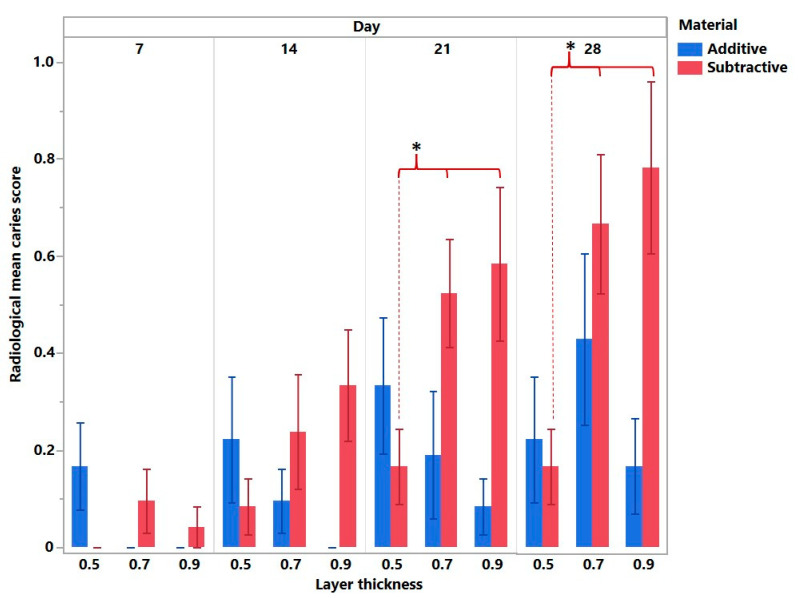
Radiological mean caries score in relation to the day of examination, material, and restoration thickness. The italics represent the statistical significance regarding the material within each day. Statistical significance with regard to restoration thickness is marked with an asterisk (marked with *).

**Figure 9 bioengineering-10-00992-f009:**
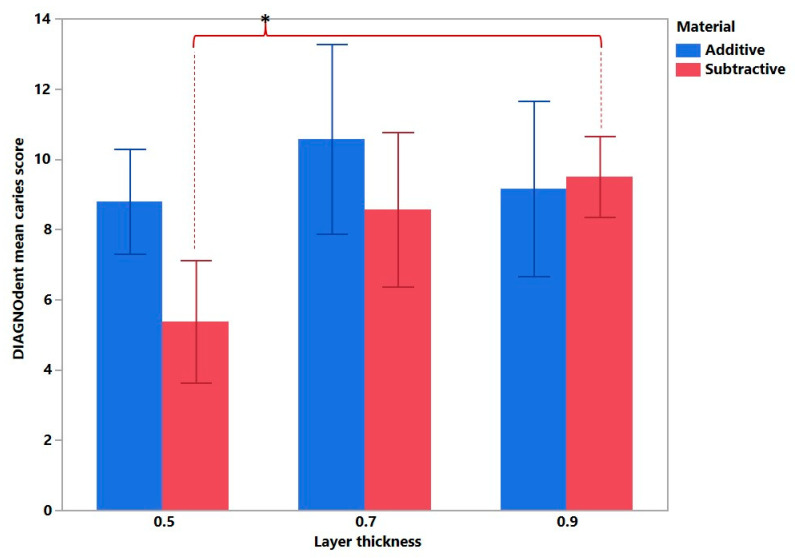
DIAGNOdent mean caries score in relation to the manufacturing method and restoration thickness. Statistical significance is marked with an asterisk. In general, a positive correlation was observed between the real caries progression (DIAGNOdent) and radiological signs of caries (Figure 10).

**Figure 10 bioengineering-10-00992-f010:**
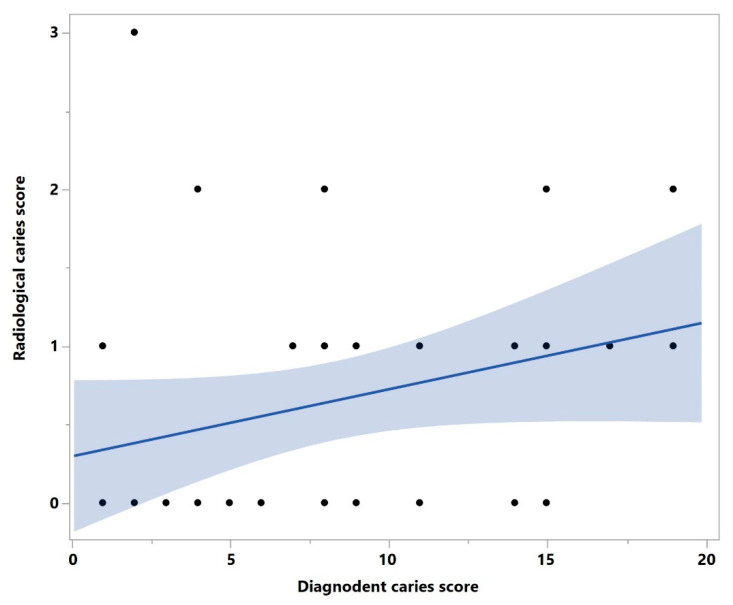
Correlation graphic between the real and radiological caries progression.

**Table 1 bioengineering-10-00992-t001:** Details of the tested CAD/CAM hybrid materials.

Material	Composition	Manufacturer	Code
VarseoSmile Crown plus	Ceramic-filled (30–50 wt% inorganic fillers; particle size 0.7 µm) silanized dental glass, methyl benzoylfor-mate, diphenyl (2, 4, 6-trimethylbenzoyl) phosphine oxide hybrid material	Bego, Bremen, Germany	VSCP
Vita Enamic	Polymer infiltrated (UDMA, TEGDMA 14 wt%) feldspar ceramic network (86 wt%)	Vita Zahnfabrik, Bad Sackingen, Germany	VE

**Table 2 bioengineering-10-00992-t002:** Radiographic scores for classification of approximal carious lesions.

Score	Description
0	No visible radiolucency
1	Radiolucency in the outer half (<half of the enamel)
2	Radiolucency in the inner half (<half of the enamel up to the enamel-dentin border)
3	Radiolucency in the dentin (broken enamel-dentin border but without obvious spread in the dentin)
4	Radiolucency with obvious spread in the outer half of the dentin (<halfway through the pulp)
5	Radiolucency with obvious spread in the inner half of the dentin (>halfway through the pulp)

## Data Availability

Raw data is available on request.

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
