# Peer review of "The Assessability of Approximal Secondary Caries of Non-Invasive 3D-Printed Veneers Depending on the Restoration Thickness—An In Vitro Study"

_bioengineering, 2023, doi:10.3390/bioengineering10090992_

Round 1

Reviewer 1 Report

General comment:

This paper investigates approximal caries development after insertion of 3D-printed, non-invasive veneers of different restoration thicknesses via the use of radiography. The manuscript provides an important insight on the potential use of radiography for the assessment of injury and would healing involving implanted 3D printed parts. There are some minor flaws that require the authors to address.

Specific comments:

1.To enhance the clarity of the experimental process, it is suggested to incorporate a schematic in the research paper, illustrating the workflow of the entire experiment. The schematic should encompass the steps from patient identification, veneer design, data collection, and any other relevant stages involved in the study.

2. In Figure 1, the authors should consider adding scale bars to each image. This addition will provide a visual reference for accurately determining the size or dimensions of the depicted features in the images, aiding readers in their understanding and interpretation of the data.

3. For improved visualization and comprehension, the researchers are advised to introduce a new figure that visually represents the different cases described in Table 2. This additional figure will help readers grasp the variations among the cases more effectively, enabling a clearer understanding of the results.

4. To convey the variability in the data accurately, it is essential to include error bars in Figure 5. As there are multiple samples for different cases, error bars will demonstrate the dispersion of the data points, providing a more comprehensive representation of the experimental outcomes.

5. The term "restoration thickness" needs to be clearly defined within the context of the research. To aid readers' understanding, it is recommended to incorporate a schematic diagram illustrating the concept of restoration thickness. This illustration will visually explain the term, making it easier for readers to comprehend its significance in the study.

6. In the introduction section, the authors should provide justification for employing X-ray as a means of investigating 3D printed parts. Additionally, it is important to highlight the distinctions between this study and previous works that have utilized X-ray for similar purposes. The researchers should reference, cite, and discuss relevant prior studies (see suggestions below) to emphasize the novelty and contributions of their research in comparison to existing literature. This approach will give readers context and insights into the unique aspects of the current study.

a. Wang, X., Zhao, L., Fuh, J. Y. H., & Lee, H. P. (2019). Effect of porosity on mechanical properties of 3D printed polymers: Experiments and micromechanical modeling based on X-ray computed tomography analysis. Polymers11(7), 1154.

b. Luis, E., Pan, H. M., Sing, S. L., Bastola, A. K., Goh, G. D., Goh, G. L., ... & Yeong, W. Y. (2019). Silicone 3D printing: process optimization, product biocompatibility, and reliability of silicone meniscus implants. 3D Printing and Additive Manufacturing6(6), 319-332.

c. Kruger, J., du Plessis, A., & van Zijl, G. (2021). An investigation into the porosity of extrusion-based 3D printed concrete. Additive Manufacturing37, 101740.

Nil.

Reviewer 2 Report

It is a very interesting study regarding secondary caries occurrence as function of 3D printed veneers thickness. Both subtractive and additive methods were used for the veneers printing with adequate commercial materials. Veneers were printed using a CAD model of the scanned teeth. The restoration were properly effectuated. Performant X-ray images were effectuated on the restored teeth revealing veneers thickness and secondary caries occurrence. DIAGNOdent was used for X-ray data confirmation.

The research was properly conducted and the obtained results are relevant, well discussed and support the drawn conclusions. However some minor aspects requires attention according to the comments below:

1) Figure 3 must be completed with X-ray images for layer thickness of 0.5 and 0.7 mm. A scale bar placed on each X-ray image would be excellent.

2) Some details regarding 3D printing regime (e.g. temperature, printing nozzle diameter, printing speed) should be added along with some photographs of the printed veneers (if you have some SEM images of the printed veneers it will be great to present them along with a short description of their microstructure).

3) Some laser fluorescence images of the formed secondary caries should be presented in order to sustain data presented in Figure 5.

4) Reference 13 is incomplete, please revise it.

5) Lines 26 and 273 ,,aquivalent" should be ,,equivalent" please revise it.

Minor editing of English language required. For instance: Lines 26 and 273 ,,aquivalent" should be ,,equivalent"
